# Curcumin Attenuates the Pathogenicity of *Entamoeba histolytica* by Regulating the Expression of Virulence Factors in an *Ex-Vivo* Model Infection

**DOI:** 10.3390/pathogens8030127

**Published:** 2019-08-15

**Authors:** Itzia Azucena Rangel-Castañeda, Pilar Carranza-Rosales, Nancy Elena Guzmán-Delgado, José Manuel Hernández-Hernández, Sirenia González-Pozos, Armando Pérez-Rangel, Araceli Castillo-Romero

**Affiliations:** 1Departamento de Fisiología, Centro Universitario de Ciencias de la Salud, Universidad de Guadalajara, Guadalajara 44340, Mexico; 2Centro de Investigación Biomédica del Noreste, Instituto Mexicano del Seguro Social, Monterrey 64720, Mexico; 3Unidad Médica de Alta Especialidad (UMAE), Hospital de Cardiología No. 34, Monterrey 64360, Mexico; 4Departamento de Biología Celular, Centro de Investigación y Estudios Avanzados del Instituto Politécnico Nacional, Ciudad de México 07360, Mexico; 5Unidad de Microscopía Electrónica LaNSE, Centro de Investigación y Estudios Avanzados del Instituto Politécnico Nacional, Ciudad de México 07360, Mexico; 6Departamento de Microbiología y Patología, Centro Universitario de Ciencias de la Salud, Universidad de Guadalajara, Guadalajara 44340, Mexico

**Keywords:** Curcumin, *Entamoeba histolytica*, Amoebic liver abscess, Virulence genes

## Abstract

Infection with the enteric protozoan *Entamoeba histolytica* is still a serious public health problem, especially in developing countries. Amoebic liver abscess (ALA) is the most common extraintestinal manifestation of the amoebiasis, and it can lead to serious and potentially life-threatening complications in some people. ALA can be cured by metronidazole (MTZ); however, because it has poor activity against luminal trophozoites, 40–60% of treated patients get repeated episodes of invasive disease and require repeated treatments that can induce resistance to MTZ, this may emerge as an important public health problem. Anti-virulence strategies that impair the virulence of pathogens are one of the novel approaches to solving the problem. In this study, we found that low doses of curcumin (10 and 50 μM) attenuate the virulence of *E. histolytica* without affecting trophozoites growth or triggering liver injury. Curcumin (CUR) decreases the expression of genes associated with *E. histolytica* virulence (*gal/galnac* lectin, *ehcp1*, *ehcp5*, and amoebapore), and is correlated with significantly lower amoebic invasion. In addition, oxidative stress is critically involved in the etiopathology of amoebic liver abscess; our results show no changes in mRNA expression levels of superoxide dismutase (SOD) and catalase (CAT) after *E. histolytica* infection, with or without CUR. This study provides clear evidence that curcumin could be an anti-virulence agent against *E. histolytica*, and makes it an attractive potential starting point for effective treatments that reduce downstream amoebic liver abscess.

## 1. Introduction

The protozoa *Entamoeba histolytica* causes several clinical syndromes, ranging from dysentery of acute colitis to amoebic liver abscess (ALA), and other extra-intestinal diseases [1,2,3]. Diverse in vitro and in vivo studies have demonstrated that multiple molecules from *E. histolytica* are involved directly or indirectly in pathogenesis. Among them: galactose/N-acetyl D-galactosamine lectin (Gal/GalNAc), the Cysteine proteinases (CPs) and the amoebapores are three well-studied virulence factors [4,5,6,7,8,9]. 

The complete sequence of morphologic events during ALA formation has been studied in vivo and in vitro models. The use of precision-cut hamster liver slices has brought remarkable contributions, from the lodgment of amebas in the hepatic sinusoids to the development of extensive liver necrosis. Importantly, the ex vivo models have contributed to use reduction of experimental animals [10,11,12].

In general, metronidazole (MTZ), tinidazole, ornidazole, and nitazoxanide are active in invaded tissues [13,14]. Due to its excellent response, the tissue agent of choice is MTZ, the cure rate for which is >90%, but because of its rapid absorption after oral administration it is not effective against luminal trophozoites; 40–60% of treated patients get repeated episodes of invasive disease and require repeated treatments that induce resistance to MTZ [15]. For these reasons, there is a significant need for new drugs or approaches to combat amoebic drug resistance. Anti-virulence strategies disarm the pathogens, thereby rendering them harmless and more susceptible to immune clearance [16,17,18]. Curcumin, the major component present in the rhizome of *Curcuma longa* L., has been extensively studied for its anti-oxidant, anticancer, anti-inflammatory, antiparasitic, and hepatoprotective properties [19,20,21,22,23,24]. Some studies have shown that low concentrations of CUR (5–50 µM) prevent hepatic injury caused by parasites and ethanol [25,26,27]. Recently, we demonstrated that 100–300 μM of CUR has amoebicidal activity, affecting *E. histolytica* trophozoites, growth, and morphology, in a dose-dependent manner [28]. In this study, we found that CUR at low concentrations (10 and 50 μM) displayed direct activity against *E. histolytica*, inhibiting the virulence factors expression and pathogenicity of this parasite; histopathological analysis and microscopy images revealed that the presence of CUR during the infection prevents liver amoebic invasion.

## 2. Results

### 2.1. Curcumin Prevents the Tissue Damage Caused by Entamoeba histolytica

To evaluate the effect of CUR during the infection of precision-cut hamster liver slices (PCHLS) by *E. histolytica*, hematoxylin and eosin staining and scanning electron microscopy assays were used. Images from H&E showed normal liver architecture in non-infected PCHLS; hepatic parenchyma, sinusoidal spaces, and central veins were observed without change (Figure 1A,B). Infected tissue slices with trophozoites showed architecture alterations, parasites invading the sinusoidal spaces; the amoebas were located close to the hepatocytes (Figure 1C,D). On the other hand, the presence of CUR during infection prevented liver tissue damage; no alterations were seen in the hepatic parenchyma, and the number of trophozoites in the sinusoidal spaces was significantly reduced (Figure 1I). With CUR 10 μM, few parasites with erythrophagocytic activity were observed in the central veins (Figure 1E,F). Moreover, CUR 50 µM-infected slices showed similar histological characteristics to non-infected liver slices (Figure 1G,H). 

To further verify effects of CUR during *E. histolytica* infection, structural tissue details were analyzed by SEM. Comparative SEM images revealed non-infected PCHLS with plates of hepatocytes around the sinusoids, and the portal tract architecture remained unaltered (Figure 2A,B). Infected PCHLS showed trophozoites on the liver parenchyma surface in close contact with hepatocytes and alterations in sinusoids, as well as disorganization of the hepatic cords (Figure 2C,D). Figure 2E–H show that the presence of 10 or 50 µM CUR during infection of PCHLS prevented tissue injury. Tissue with normal characteristics, and intact histological relationship between the sinusoidal spaces and the hepatocyte plates, was observed. 

### 2.2. Effect of Curcumin on PCHLS Viability

The cell viability of infected and CUR-infected PCHLS was determined using the reduction of the metabolic indicator resazurin. After 18 h of infection, the level of reduction of resazurin suggested a significant inhibition of cell viability (23.8%) by *E. histolytica*, compared with uninfected slices. With CUR 10 µM-infected, the viability of PCHSL was inhibited only 12%. Interestingly, there was no significant differences between CUR 50 µM-infected and uninfected PCHLS (Figure 3).

### 2.3. Effect of Curcumin on Antioxidant Enzymes Expression

Oxidative stress is critically involved in the etiopathology of ALA [29]. There has been evidence that CUR exerts a hepatoprotective effect against oxidative stress via up-regulation of antioxidant enzyme genes [30,31]. The SOD and CAT mRNA levels in the PCHLS were determined to evaluate the impact of CUR during the amoebic infection. Results showed no significant difference in the expression between infected and CUR-infected slices (Figure 4).

### 2.4. Effect of Curcumin on the Growth of Entamoeba histolytica

In studies conducted to determine whether CUR can inhibit the growth of *E. histolytica* in vitro. CUR was found that at 10 or 50 μM has null activity against trophozoites (Figure 5).

### 2.5. Curcumin Regulates the Expression of Virulence Factors

To determine whether low concentrations of CUR had a direct effect on parasite virulence during PCHLS amoebic infections, we measured the expression levels of *gal/galnac* lectin, *ehcp1*, *ehcp5,* and amoebapore mRNA by semiquantitative RT-PCR in infected PCHLS (Figure 6). After 18 h of amoebic infection, there were no statistically significant differences in the expression level of virulence factors between uninfected and CUR 10 µM-infected slices. However, the presence of CUR 50 µM on infected slices, interestingly, down-regulated mRNA levels of *gal/galnac* lectin, *ehcp1*, *ehcp5*, and amoebapore. The most dramatic impact on gene expression was observed for *gal/galnac* lectin (90%) and *ehcp1* (89.6%). The effect of CUR on expression levels of virulence factors also was evaluated in vitro, and no difference was found with the results obtained in the PCHLS (data no shown). 

## 3. Discussion

*E. histolytica* has been associated with around 100 million cases of amoebic dysentery, colitis, and ALA that lead to almost 50,000 deaths each year [32,33]. Virulence factors of the pathogen and specific and nonspecific host defense mechanisms are key factors in determining outcome to infection [34]. CUR, the main constituent of turmeric, has demonstrated therapeutic potential to control of parasitic and liver diseases [35]. Recently, we reported the amoebicidal effect of CUR against *E. histolytica* [28]. However, the mechanism of action has not been elucidated. In this study, we describe the role of CUR on the expression of the major virulence factors of this parasite and on liver damage. Using an ex vivo model to induce the formation of ALA, we demonstrate destruction of liver tissue by the parasite. Similar to Carranza and co-worker results [36,37], our histological and SEM analysis of infected PCHLS showed changes in the tissue architecture with trophozoites in close contact with hepatocytes, especially on the fenestrated surface of the endothelium that covers these cells. On the other hand, some authors have reported the hepatoprotective properties of CUR [26,27,35,38]; in this study, we found that during the amoebic infection the tissue was well preserved by the presence of this phytochemical, and the liver conserved its normal architecture. Furthermore, we observed a portal space with characteristic hepatic artery and portal vein. The presence of parasites in infected PHCLS significantly decreased with CUR 10 µM, and with 50 µM of CUR no parasites were observed, which correlates with the viability assay where, although the slices were infected in the presence of CUR 10 µM, the viability cell was reduced by 12.6 %, while no significant difference was detected between uninfected and CUR 50 µM-infected-PCHLS. Oxidative stress is critically involved in the etiopathology of ALA; SOD and CAT mRNA levels are upregulated during the amoebic liver infection in hamster [29]. Our observations confirmed a high-level expression of both enzymes in amoebic-infected PHCLS. In addition, several studies have shown that CUR exerts remarkable protection against oxidative stress associated liver diseases, through various cellular and molecular mechanisms. Those mechanisms include ameliorating cellular responses to oxidative stress such as the expression of SOD, CAT, and others [26,27,35,38]. In contrast to this evidence, in amoebic liver damage the presence of CUR during infection did not modify the expression levels of these genes (Figure 4), suggesting that CUR prevents ALA by a mechanism that does not involve its antioxidant properties.

Diverse in vitro studies have demonstrated that the principal virulence factors involved directly or indirectly in the pathogenesis of *E. histolytica* are the GalNAc lectin, the CPs, and the amoebapores [5,7,8,9]. In our infected-PCHLS, the expression of Gal/GalNAc was upregulated, which is consistent with a previous report that shows that Gal/GalNAc expression is upregulated steadily following host–parasite interaction [5]. Therefore, we investigated the effect of CUR in the mRNA expression of the most studied *E. histolytica* virulence factors, and our evidence demonstrated that all virulence factors tested were down regulated by CUR. With CUR 50 µM, *gal/galnac* and *ehcp1* expression was inhibited more than 89%. The same trend was observed when trophozoites were incubated with 10 or 50 μM of CUR. This explains why the trophozoites in the presence of CUR were not able to invade liver slices. All our findings indicate that CUR has direct activity against *E. histolytica*, and attenuates the pathogenicity of this parasite during amoebic liver infection. These findings are further supported by our in vitro findings that show that low doses of CUR did not affect the trophozoites growth. Previous studies have demonstrated that the inhibition of cysteine proteinase expression significantly decreased the formation of liver lesions [39,40]. In addition, in bacteria has been described that CUR attenuates its virulence [41,42,43]. In *Pseudomonas aeruginosa*, treatment with CUR downregulates virulence factors (pyocyanin, elastase, and protease) and inhibits the adherence of the bacteria to polypropylene surfaces [41,43]. This study provides clear evidence that CUR reduces the pathogenicity of *Entamoeba histolytica* by regulating the expression of virulence required to the processes of tissue invasion, suggests that this polyphenol could be an anti-virulence agent against *Entamoeba histolytica*, and makes it an attractive potential starting point for effective treatments that reduce downstream ALA.

## 4. Materials and Methods

### 4.1. Ethics Statement

The animal management protocols were approved by the institutional committee (C.I. 010/15). All the procedures were carried out following federal regulations for the production, care, and use of laboratory animals (NOM-062-Z00-1999).

### 4.2. Entamoeba histolytica Trophozoites Culture

Virulent trophozoites of *E. histolytica* strain HM1-IMSS were grown axenically at 37 °C in PEHPS media [44] and supplemented with 10% bovine serum. Cultures were routinely maintained by amoeba sub-culturing twice a week. The virulence was maintained by routinely direct inoculation of trophozoites into liver hamster. For in vitro experiments, trophozoites were harvested in the log phase growth. 

### 4.3. Precision-Cut Hamster Liver Slices Preparation

Precision-cut hamster liver slices were prepared as described previously [36]. Briefly, 2 months old male Syrian golden hamster (*Mesocricetus auratus*) were sacrificed with sodium pentobarbital (6 mg/100 g), then the liver was removed and placed into Krebs buffer. The hepatic lobes were cut into cylinders of 10 mm diameter and sliced into 200–250 µm thick tissue slices with a Brendel Vitron tissue slicer (Vitron, Tucson, AZ, USA) in oxygenated KB buffer (95:5% O_2_:CO_2_). After, liver slices were placed in 24-well polystyrene microplates with 1 mL of DMEM/F12 media and incubated 1 h at 37 °C to stabilize the tissue before infection.

### 4.4. Preparation of Curcumin Stocks

The CUR was obtained from Sigma Chemicals Co. (St Louis, MO, USA). The stock of CUR was prepared to a concentration of 40 mg/mL, dissolved in dimethyl sulfoxide (DMSO), and diluted with media to a concentration of 10 and 50 µM.

### 4.5. Ex Vivo Infection

For infection experiments, an inoculum of 100,000 trophozoites per slice, resuspended in 1 mL of DMEM/F12:PEHPS media (1:1), in a 24-well plate was used. The culture media was supplemented with 2.24 g/L sodium bicarbonate, 50 µg/mL gentamicin, 25 mM glucose, 1% insulin-transferrin-selenium mix (Sigma-Aldrich), and 10% fetal bovine serum. At the same time, PCHLS were exposed to 10 and 50 µM of CUR. Slices without CUR were used as control. First, the plates were incubated for 2 h at 37 °C to allow the interaction and penetration of the trophozoites into the tissue, and then incubated for 18 h at 37 °C with slow agitation (25 rpm) to allow the formation of liver abscess. Finally, the PCHLS were processed for histopathological, scanning electron microcopy (SEM), and molecular analyses. 

### 4.6. Histopathological Image Analysis

To evaluate whether CUR reduced the histopathological damage caused by *E. histolytica* in PCHLS, uninfected and infected tissue slices were fixed in 10 % neutral formalin for 24 h and embedded in paraffin. For the study, 5 μm tissue sections from each slice were placed in xylene, rehydrated using a graded ethanol (100–75%) (each time for 3 minutes and rinsed with water). Later, the slices were stained with hematoxylin for 10 minutes, rinsed with water, and then stained with eosin. Finally, the tissue sections were dehydrated in a graded ethanol series (75–100%) and examined by light microscopy. All photomicrographs were obtained using a Zeiss Axiostar Plus Brightfield microscope. 

### 4.7. Scanning Electron Microscopy

The morphological changes caused by *E. histolytica* infection in PCHLS were also analyzed by SEM. The uninfected and infected PCHLS were fixed with 2.5% glutaraldehyde in phosphate buffered saline (PBS). The PCHLS were washed three times with PBS and post-fixed for 2 h in 2% osmium tetroxide. Next, the fixed PCHLS were washed with PBS, dehydrated in a graded ethanol series (50–100%), and subjected to critical-point drying with CO_2_. Finally, the slices were sputter-coated with gold and examined in a JEOL-JSM6510LV SEM (JEOL, Japan).

### 4.8. Viability Assay

Resazurin assay was used to determine the cell viability/metabolic activity [45]. Briefly, uninfected and infected PCHLS and exposed to CUR 10 or 50 μM were placed in a 24-well plate, washed with PBS once, and incubated at 37 °C for 4 h with slow agitation (25 rpm) in DMEM medium containing 10% resazurin reagent (0.1 mg/mL) (Sigma). After incubation time, fluorescence intensity was measured in a microplate reader (Synergy HT, BioTek) at 530 nm excitation/590 nm emission wavelengths. Medium without resazurin and completely reduced resazurin were used as controls. The quantitative analysis of viability was determinated using the alamarBlue® Fluorometric Calculator (https://www.bio-rad-antibodies.com/colorimetric-calculator-fluorometric-alamarblue.html#).

### 4.9. Expression of Virulence Factors and Antioxidant Enzymes in PCHLS

The effect of CUR on the expression of genes coding for antioxidant enzymes superoxide dismutase (SOD) and catalase (CAT) in hamsters and genes associated with the virulence of *E. histolytica* (*gal/galnac* lectin, *ehcp1*, *ehcp5*, and amoebapore) was evaluated by semi-quantitative RT-PCR. Total RNA from uninfected and infected PCHLS and incubated along with CUR was isolated using total RNA purification kit (NORGEN). The quantity and purity were measured using a biophotometer (Eppendorf D30). The RNA samples (1 μg) were reverse transcribed to cDNA using SuperScript III Reverse Transcriptase (INVITROGEN) and oligo dT primer, according to the manufacturer’s instructions. For RT PCR, the oligonucleotides sequences used in this study are summarized in Table 1. Actin and glyceraldehyde-3-phosphate dehydrogenase (*gpdh*) were used as endogenous control genes. The RT-PCR included an initial denaturation at 98 °C for 30 s and 40 cycles of 98 °C for 10 s; 54 °C (*gal/galnac*, *sod*, *cat* and *gpdh*), 60 °C (*ehcp1*, *amoebapore* and *actin*), or 65 °C (*ehcp5*) for 30 s; and 72 °C for 1 min. A final extension of 7 min at 72 °C was performed. The amplification of each gene was performed by triplicate, and its differential expression calculated through normalization against a reference gene.

### 4.10. Growth Inhibition Assay

To evaluate the effect of CUR on *Entamoeba histolytica* growth, an inoculum of 15,000 parasites/mL was grown in the presence of 10 or 50 μM of CUR (Sigma-Aldrich) in TYI-S-33 medium at 37 °C for 12, 24 and 48 h. The diluent of CUR, 0.5% DMSO (Sigma-Aldrich) was used as negative control. After incubation periods, the cells were harvested by cooling and counted using a Neubauer chamber. The percentage of parasite growth inhibition was calculated in relation to negative control, which was defined as 100% parasite growth development. Experiments were performed by triplicate and repeated three different times.

### 4.11. Statistical Analysis

Densitometric analysis of gene expression levels was performed with Image J analysis software (NIH). All data were presented as mean values with standard deviations and analyzed using two-way ANOVA followed by Dunnett’s multiple comparisons test (GraphPad Prism version 6.01 for Windows, GraphPad Software, La Jolla California USA). *p* values of ≤ 0.05 were considered significantly different.

## Figures and Tables

**Figure 1 pathogens-08-00127-f001:**
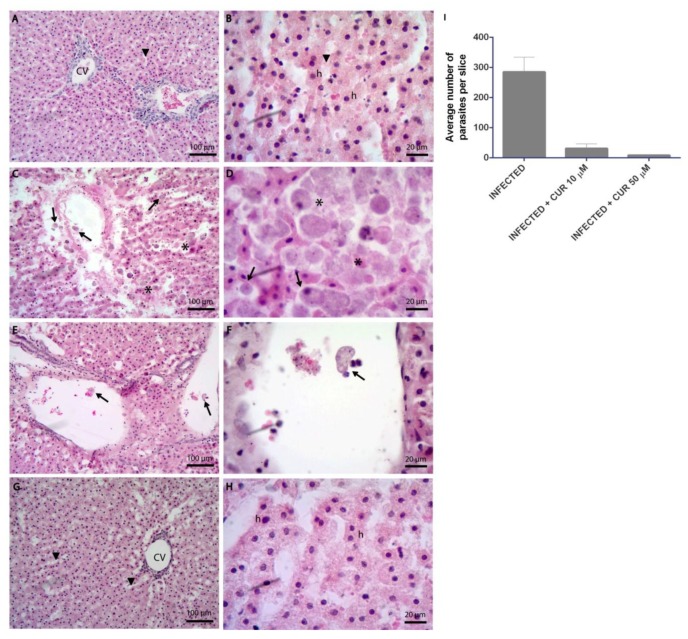
Representative histological views of H&E staining of precision-cut hamster liver slices (PCHLS) showed liver injury and *Entamoeba histolytica* invasion. (**A**,**B**) Non-infected. (**C**,**D**) Infected. (**E**,**F**) CUR 10 µM-infected. (**G**,**H**) CUR 50 µM-infected. CV = central vein and h = hepatocyte. Arrows indicate trophozoites, asterisks denote parasite aggregation, and arrowheads indicate the sinusoidal spaces. Images are representative of five independent experiments. (**I**) Average number of parasites in PCHLS, estimated by counting trophozoites in a series of three independent slices.

**Figure 2 pathogens-08-00127-f002:**
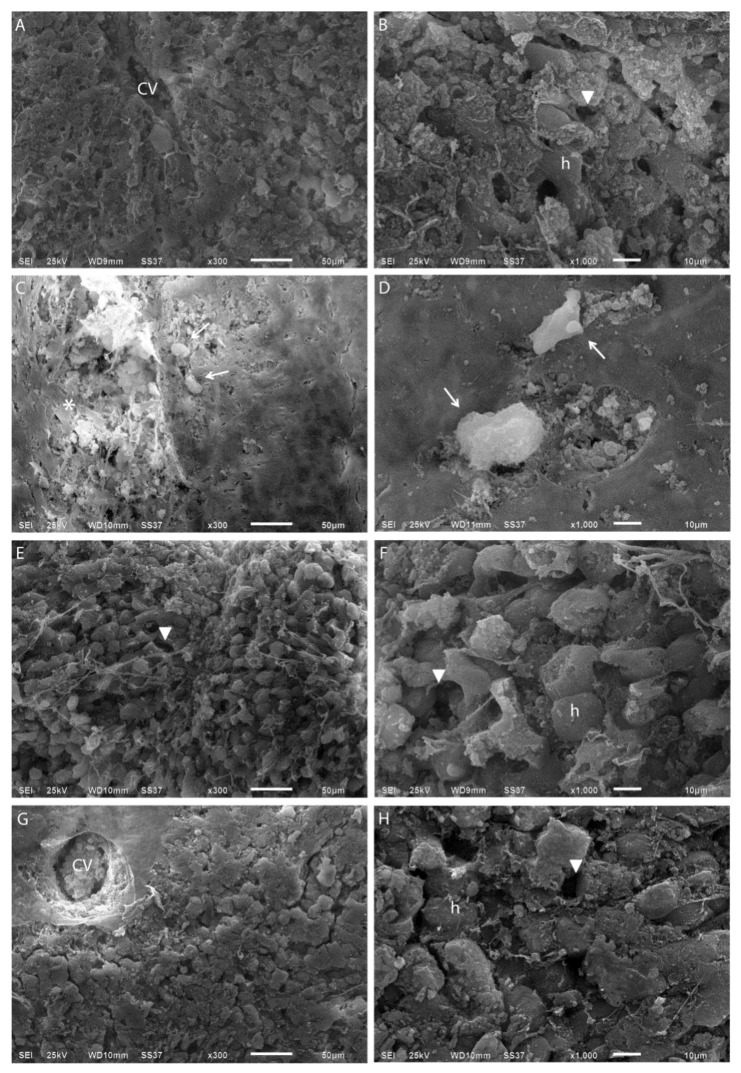
Comparative scanning electron microscopy micrographs of PCHLS. (**A**,**B**) Non-infected PCHLS. (**C**,**D**) Infected. (**E**,**F**) CUR 10 µM-infected PCHLS. (**G**,**H**) CUR 50 µM-infected PCHLS. CV = central vein and h = hepatocyte. Arrow heads denote sinusoidal spaces, arrows indicate trophozoites, and asterisks denote disorganization of the hepatic cord.

**Figure 3 pathogens-08-00127-f003:**
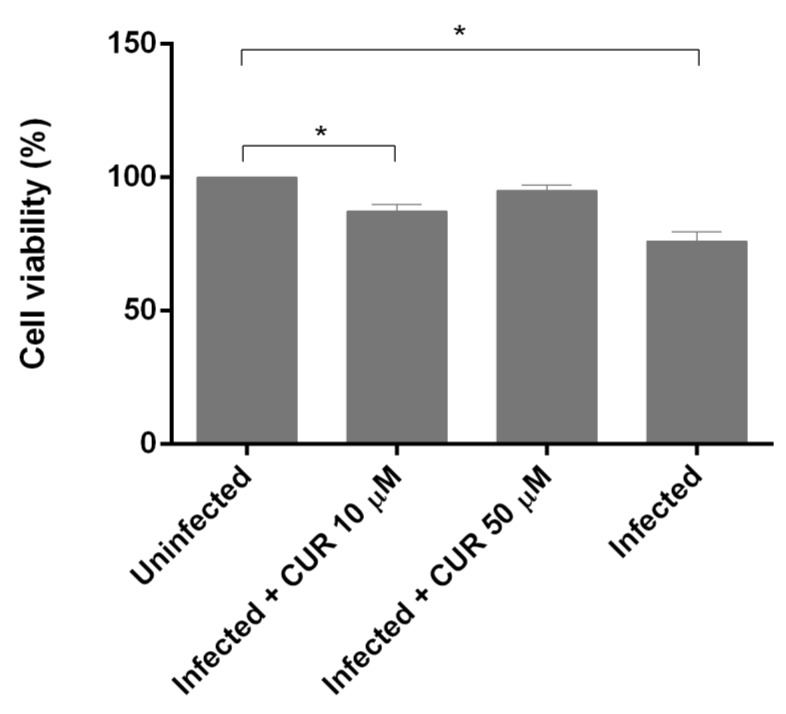
Effect of curcumin on viability of PCHLS infected with *E. histolytica.* Data correspond to mean values ± SD. * *p* < 0.05.

**Figure 4 pathogens-08-00127-f004:**
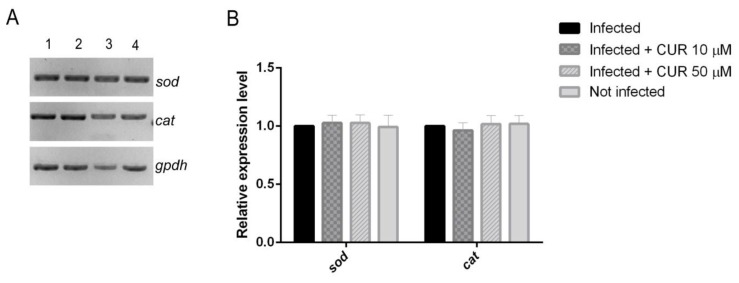
Effect of CUR on the mRNA expression levels of *sod* and *cat* in PCHLS. (**A**) Representative gel image. Lane 1, infected; lane 2, infected + CUR 10 µM; lane 3, infected + CUR 50 µM; lane 4, not infected. (**B**) Graphical representation of the densitometric analysis. Data correspond to mean values ± SD of three independent experiments (Appendix A).

**Figure 5 pathogens-08-00127-f005:**
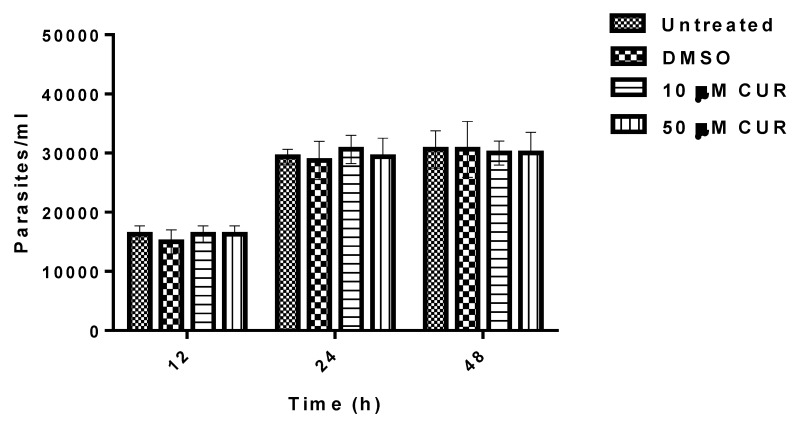
Effect of curcumin on the growth of *E. histolytica*. Data correspond to mean values ± SD of three independent experiments.

**Figure 6 pathogens-08-00127-f006:**
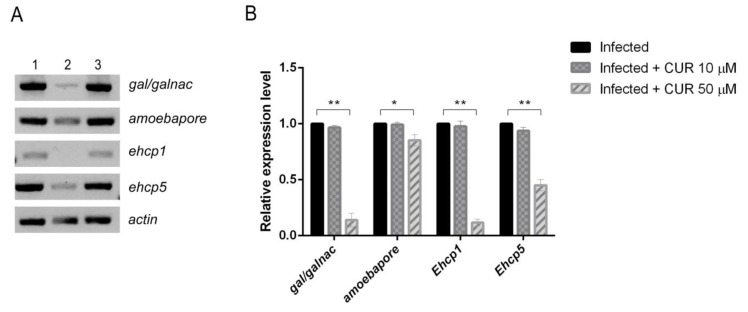
Effect of CUR on relative expression of *gal/galnac*, *ehcp1*, *ehcp5* and *amoebapore*. (**A**) Representative gel image. Lane 1, infected; lane 2, infected + CUR 50 µM; lane 3, infected + CUR 10 µM. (B) Graphical representation of densitometric analysis. Data correspond to mean values ± SD of three independent experiments (Appendix A). * *p* < 0.05 and ** *p* < 0.001.

**Table 1 pathogens-08-00127-t001:** List of oligonucleotides used.

Gene	Gen Bank Accession	Sequence (5ʹ-3ʹ)	Amplicon Size (bp)
*sod*	XM_005073823	AGGGCACCATCCACTTCGAGCAG	224
		AGCAGTCACGTTGCCCAGGT	
*cat*	XM_005064849	GCAGCTCAGAAATCCTACACCA	332
		CCAATGTGCTCAAACACCTTTGC	
*gpdh*	U10983	CCAACGTGTCCGTCGTGGAT	255
		ACCCTGTTGCTGTAGCCGAA	
*ehcp 1*	XM_645064	CATGTAGAAGTGATGTGAAAGC	247
		TTCTTTCCCATCAACAACA	
*ehcp 5*	XM_645845	TCCAGCTATTAGAGACCAAGCATC	398
		TAACTCCAGAAGCATCAATAGC	
*amoebapore*	M83945	AAGGAGAAATCCTCTGCAAC	216
		CAAATAGCATTGGCATCAAC	
*gal/galnac*	XM_651089	CACTTGTCAAATACACAGCAGGAC	484
		GGTTTAGCTTTAGGCCATGGAA	
*actin*	XM_001914602	AATGAAAGATTCAGATGCCC	283
		ATTGATCCTCCAATCCAGAC	

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
