# Peer review of "Curcumin Attenuates the Pathogenicity of Entamoeba histolytica by Regulating the Expression of Virulence Factors in an Ex-Vivo Model Infection"

_pathogens, 2019, doi:10.3390/pathogens8030127_

Round 1
Reviewer 1 Report
Authors present a study on the effect of curcumin on the pathogenicity of Entamoeba histolytica.
Even though it is shown that curcumin might act as an anti-virulence agent against E. histolytica, the manuscript presents aspects that should be improved.
Introduction:
Line 59 - Other treatments are available. They should be included and discussed.
Results:
Fig. 4: Two indendent experiments is a very limited number to perform a strong statistical analysis. At least 3 biological replicates are considered necessary to confirm the effects.
Gel/blots images should be revised. Full-length gels or blots have to be delivery (could be included in supplementary data) in order to comply with the most recent integrity policies.
If gels are cropped from different parts of the same or from different gels, or different exposures are used, all must be made explicit in the manuscript and legende of the figure.
Fig. 5: Again 2 experiments is too limited.
Fig 6. Same remarks for the gels. Please increase n.
Minor remarks:
Line 211: correct O2 and CO2
Author Response
Response to Reviewer 1 Comments
Point 1: Line 59- Other treatments are available. They should be included and discussed.
Response 1: We thank Reviewer 1 for bringing this to our attention. We annexed to the manuscript a short description of the others ALA treatments. See lines 59-61.
Point 2: Two independent experiments is very limited number to perform a strong statistical analysis. At least 3 biological replicates are considered necessary to confirm the effects.
Gel/blots images should be revised. Full-length gels or blots have to be delivery (could be included in supplementary data) in order to comply with the most recent integrity policies.
If gels are cropped from different parts of the same or from different gels, or different exposures are used, all must be made explicit in the manuscript and legend of the figure.
Response 2: A new experiment was performed to complete the three replicates. The blots were cropped from same gel and the images 4 and 6 were modified. See representative image of Full-length gels in the attached file.
Point 3: Figure 5, again 2 experiments are too limited.
Response 3: A new experiment was performed to complete the three replicates. The figure 5 was modified.
Point 4: Figure 6, same remarks for the gels. Please increase n
Response 4: Answered in point 2.
Point 5: Line 211, correct O2 and CO2.
Response 5: Corrected in the text, see line 211.

Reviewer 2 Report
Curcumin attenuates the pathogenicity of E. histolytica by regulating the expression of virulence factors in an ex-vivo model infection.
In this work, Rangel-Castañeda et al., Try to demonstrate that curcumin, through its antiparasitic properties, protects slices of hamster liver against the pathogenicity of E histolytica, also, describes that curcumin has an inhibitory effect on the genes of some virulence factors of the amoeba.
From this work we can only consider the in vitro experiments of the interaction of Eh trophozoites and curcumin on their antiparasitic effects that the authors have published and the inhibitory effect of in vitro virulence factor genes, since the model of Liver slices lack all the blood circulation factors such as cells and complement that have been shown to participate in amoebic liver injury such as neutrophils and macrophages, thus, it can not be asserted that curcumin has a hepatoprotective effect, as there are no such cells that are essential in the inflammatory phenomenon when carrying out the amoebic liver abscess model in the whole animal, from the histopathological point of view, it does not contribute much to the images of HE and SEM, since the damage to the hepatocytes is not appreciated, there is no necrosis, balonization, edema, this damage could be better appreciated if fine histological sections were developed s to be observed under the transmission electron microscope.
To complete the work, I suggest that the authors:
1.- Experimenting in triplicate so that the results have an adequate statistical significance.
2.- Demonstrate the presence of trophozoites in liver slices treated and not treated with curcumin.
3.- Count the trophozoites present in all the phases of the experiments, and make graphs of those results.
4.- Discuss why in the slices with trophozoites and curcumin 50 uM there are no trophozoites, and if this does not cause that by means of semiquantitative RT-PCR a drastic decrease in the pathogenicity factors of gal / galnac lectin, ehcp1, ehcp5 and amoebapore is observed ?
Minor things:
1.- Bibliographic citation 1 is not related to amoebic complications that are mentioned in lines 48 and 49 of the introduction. It is still necessary to review the article by Espinoza Cantellano and MartÃnez-Palomo 2000.
2.- There are no bibliographical citations that support the description of the clinical complications of amoebiasis mentioned in the introduction (page 3, lines 48 and 49).
3.- Citation 3 (page 3, line 53) does not refer to amoebic molecules.
4.- There are abbreviations that are not described in their first mention (PCHLS)
5.- Because curcumin does not affect growth in vitro, and in tissues, trophozoites are not appreciated, especially at a dose of 50 uM?. Explain.
Author Response
Response to Reviewer 2 Comments
Point 1: Experimenting in triplicate so that the results have an adequate statistical significance.
Response 1: We thank Reviewer 2 for bringing this to our attention. A new experiments were performed to complete the three replicates.
Point 2: Demonstrate the presence of trophozoites in liver slices treated and not treated with curcumin.
Response 2: We do not understand this question, our H&E and SEM images shown the trophozoites presence in liver slices treated and not treated with curcumin.
Point 3: Count the trophozoites present in all the phases of the experiments, and make graphs of those results.
Response 3:
Our H&E and SEM results are representative images, in infected slices without treatment, hepatic invasion by amebic trophozoites results in marked tissue destruction, and due to the invasive presence of trophozoites it is difficult to account the total number of trophozoites. See representative images of infected slices in the attached file.
Point 4: Discuss why in the slices with trophozoites and curcumin 50 uM there are no trophozoites, and if this does not cause that by means of semiquantitative RT-PCR a drastic decrease in the pathogenicity factors of gal / galnac lectin, ehcp1, ehcp5 and amoebapore is observed ?
Response 4: Our manuscript now included a short description of CUR on invasion prevention. See lines 85-87.
Point 5: Bibliographic citation 1 is not related to amoebic complications that are mentioned in lines 48 and 49 of the introduction. It is still necessary to review the article by Espinoza Cantellano and MartÃnez-Palomo 2000.
Response 5: The citation 1 has been modified and the article by Espinoza Cantellano and MartÃnez-Palomo 2000 was included. See line 49.
Point 6: There are no bibliographical citations that support the description of the clinical complications of amoebiasis mentioned in the introduction (page 3, lines 48 and 49).
Response 6: Corrected in the text, see line 49.
Point 7: Citation 3 (page 3, line 53) does not refer to amoebic molecules.
Response 7: The citation 3 has been eliminated.
Point 8: There are abbreviations that are not described in their first mention (PCHLS).
Response 8: Corrected in the text, see line 79.
Point 9: Because curcumin does, and in tissues, trophozoites are not appreciated, especially at a dose of 50 µM. Explain.
Response 9: The treatment with curcumin at a dose of 50 µM not affect growth in vitro, but reduces capacity of Entamoeba histolytica to invade liver tissue by regulating the expression of virulence factors.

Round 2
Reviewer 1 Report
Authors have improved the manuscript as recommended. Blots presented in the rebuttal letter must to be included as supplementary data of the manuscript, together with the blots of the other 2 replicates. A representative blot is only accepted if all the blots are shown.
Figures 4A and 6A are now elucidative of the representative situation. However, Figures 4B and 6B are still the ones presented in the previous version (n=2). Please include the result of the 3 replicates in the graphics 4B and 6B.
Author Response
Response to Reviewer 1 Comments
Point 1: Authors have improved the manuscript as recommended. Blots presented in the rebuttal letter must to be included as supplementary data of the manuscript, together with the blots of the other 2 replicates. A representative blot is only accepted if all the blots are shown.
Response 1: We thank Reviewer 1 for bringing this to our attention. The blots have been included in the supplementary data.
Point 2: Figures 4A and 6A are now elucidative of the representative situation. However, Figures 4B and 6B are still the ones presented in the previous version (n=2). Please include the result of the 3 replicates in the graphics 4B and 6B.
Response 2: The result of the 3 replicates in the graphics 4B and 6B has been included. See line 123 and 145.
Reviewer 2 Report
the manuscript could be improved if authors count trophozoites in all experimental slices, control included, and adding a graph
Author Response
Response to Reviewer 2 Comments
Point 1: The manuscript could be improved if authors count trophozoites in all experimental slices, control included, and adding a graph.
Response 1: We thank Reviewer 2 for bringing this to our attention. Our revised manuscript included a graph showing the average of number of trophozoites of three slices from each treatment. Please see the Figure 1.
Round 3
Reviewer 1 Report
Authors improved the manuscript that can now be accepted for publication.